# Repair of oxidized methionine residues in the chaperone Spy maintains periplasmic proteostasis under chlorite stress in *Escherichia coli*

Laurent Loiseau[1], Nathan De Visch[1], Alexandra Vergnes[1], Jean Armengaud[2], Maxence S. Vincent[1], Benjamin Ezraty[1]*

1 Laboratoire de Chimie Bactérienne, Institut de Microbiologie de la Méditerranée, Aix-Marseille University, CNRS, Marseille, France, 2 Département Médicaments et Technologies pour la Santé, Université Paris-Saclay, CEA, INRAE, SPI, Bagnols-sur-Cèze, France

* ezraty@imm.cnrs.fr

## Abstract

The bacterial cell envelope is exposed to various stresses, including oxidative stress caused by different types of oxidants, such as reactive oxygen species (ROS) and reactive chlorine species (RCS). In Escherichia *coli,* the reduction of chlorate into chlorite, a toxic RCS compound, induces the expression of the MsrPQ system, which repairs periplasmic proteins oxidized at methionine residues (methionine sulfoxide, Met-O). In this study, using a proteomic-based approach, we show that chlorite stress also triggers the overproduction of the periplasmic molecular chaperone Spheroplast Protein Y (Spy). This response is mediated by the activation of the BaeSR two-component system. Furthermore, both in vivo and in vitro evidence reveal that Spy's susceptibility to oxidation is critical for its chaperone activity. We demonstrate that the MsrPQ repair system ensures Spy's functionality by reducing its Met-O, thereby safe-guarding its role in periplasmic protein homeostasis. Overall, this work reveals Spy as a key target of chlorite-induced oxidative damage and underscores the essential role of MsrPQ in preserving periplasmic protein quality control.

## Introduction

Preserving protein homeostasis (proteostasis) and ensuring protein functionality are crucial processes for cells [1]. In bacteria, the cell envelope in general and the peri-plasm of gram-negative bacteria in particular, are exposed to endogenous and exog-enous oxidative stresses that damage proteins [2,3]. The accumulation of damaged proteins disrupts important biological processes and can result in cellular dysfunction and even cell death. To address this damage and maintain proteostasis, bacteria rely on a combination of oxidative species-scavenging enzymes, molecular chaperones, proteases, and repair enzymes [4,5].

of the Creative Commons Attribution License, which permits unrestricted use, distribution, and reproduction in any medium, provided the original author and source are credited.

**Data availability statement:** The mass spectrometry proteomics data have been deposited to the ProteomeXchange Consortium via the PRIDE partner repository with the dataset identifier PXD068073, PXD068145, and PXD068189.

**Funding:** This work received supported from: Agence Nationale de la Recherche (#ANR-16-CE11-0012-02 METOXIC) to B.E. Agence Nationale de la Recherche—France 2030 Program (#ANR-24-INBS-0015 PROFI) for contributing to the mass spectrometry upgrade of the ProGénoMix platform to J.A. Horizon Europe Framework Programme—Marie Skłodowska-Curie MSCA Postdoctoral Fellowship (#101106503 MOR-AGE) to M.S.V. Initiative d'Excellence d'Aix-Marseille Université—A*MIDEX, Institute of Microbiology, Bioenergies and Biotechnology (IM2B) (#AO-IM2B-NE-2024-02-VINCENT) to M.S.V. The funders had no role in study design, data collection and analysis, decision to publish, or preparation of the manuscript.

**Competing interests:** The authors have declared that no competing interests exist.

**Abbreviations:** BV, benzyl viologen; $ClO_2^-$, chlorite; Cys, cysteine; DDA, data-dependent acquisition; Dsb, disulfide bond; Im7, immunity protein 7; Met, methionine; Met-O, methionine sulfoxide; Msr, methionine sulfoxide reductase; PBS, phosphate-buffered saline; PCR, polymerase chain reaction; RCS, reactive chlorine species; ROS, reactive oxygen species; Spy, Spheroplast Protein Y; α-LA, α-lactalbumin.

In the periplasm, where ATP is absent, folding and degradation factors operate through ATP-independent mechanisms. In *Escherichia coli*, a network of periplasmic molecular chaperones and proteases has been identified [5,6], including the protease/chaperone DegP, the prolyl isomerase FkpA, the chaperones SurA and Skp, which assist in outer membrane insertion, and the chaperone Spy (Spheroplast Protein Y). Spy is a 16 kDa chaperone that prevents protein aggregation and promotes client protein refolding at sub-stoichiometric concentrations [7]. Spy's active form is a cradle-shaped dimer containing a large hydrophobic surface at its core and basic residues on its edges [8,9]. The immunity protein 7 (Im7) serves as a model client for Spy [8], enabling investigation of the mechanism of chaperone action and demonstrating that Spy uses complementary charge interactions to bind to its unfolded substrates [8,10]. The expression of the *spy* gene is regulated by the Cpx and Bae envelope stress pathways [7], resulting in its upregulation during stress conditions such as exposure to ethanol, copper, and tannins [7,11].

The periplasm is more oxidizing than the cytoplasm and is the first compartment to encounter exogenous oxidizing agents. Consequently, reduction enzymes targeting the highly susceptible sulfur-containing amino acids cysteine (Cys) and methionine (Met)—specifically the Dsb (disulfide bond) and Msr (methionine sulfoxide reductase) enzyme families—are essential components of periplasmic protein quality control [3]. MsrPQ, the periplasmic Msr system widely conserved in gram-negative bacteria, reduces protein-bound methionine sulfoxide (Met-O) [12]. In *E. coli*, the expression of the *msrPQ* operon is induced by reactive chlorine species (RCS), such as HOCl (bleach), *N*-chlorotaurine, and chlorite ($ClO_2^-$), via the HprSR pathway [12–14]. The toxic oxidizing agent chlorite, produced by the reduction of chlorate ($ClO_3^-$) by nitrate reductases, oxidizes Met in periplasmic proteins, supporting the role of MsrPQ as an anti-chlorite defense system [14].

Here, we identify the Spy molecular chaperone as a key player during chlorite treatment. The Bae pathway appears to be activated by chlorite stress, leading to the overproduction of Spy. We found that Spy is oxidized under chlorite stress, which is reversible through the action of MsrPQ. Further analyses revealed that the oxidation of Met residues in the cradle region disrupts Spy's chaperone activity. Overall, we propose a model in which MsrPQ plays a crucial role in preserving chaperone function in the periplasm, highlighting its importance in periplasmic proteostasis.

## Results

### Exposure to chlorite induces Spy synthesis

In a previous study [14], we found that chlorite induces the expression of the *hiuH-msrPQ* operon. We investigated whether other proteins were overproduced in *E. coli* during chlorite stress. To this end, we performed a global label-free proteomic analysis of cells, recording 798,574 MS/MS spectra, identifying 15,818 peptide sequences, and monitoring 1,496 proteins. Comparative proteomics revealed that Spy, a periplasmic molecular chaperone, was the most overproduced protein in response to chlorite, with a fold change of 15.3 (Fig 1A and S1 Data). Consistent with

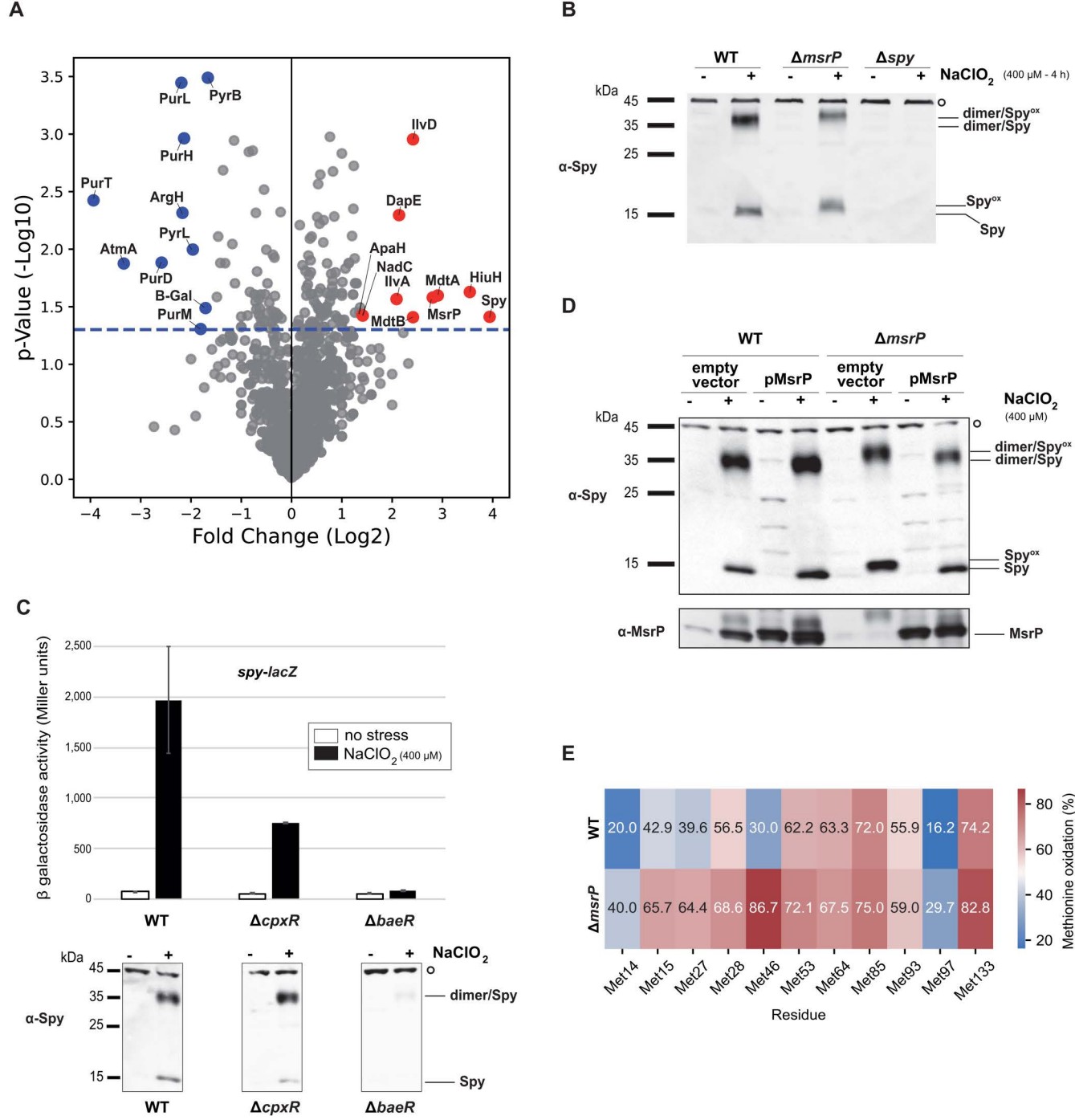

**Fig 1. Chlorite exposure induces Spy production and reversible oxidation. A.** Volcano plot of mass spectrometry results showing proteins significantly upregulated by chlorite exposure. The 10 proteins with the greatest fold changes are highlighted in red, and the 10 smallest ones in blue. The names of the corresponding proteins are indicated in the figure. The horizontal dashed line indicates the *p*-value threshold. The data underlying this panel can be found in S1 Data, and have also been deposited to the ProteomeXchange Consortium via the PRIDE partner repository with the dataset identifier PXD068145 and 10.6019/PXD068145. **B.** Immunodetection of Spy in cell extracts from WT, Δ*msrP* (CH380), and Δ*spy* (LL1392) strains. Spy levels were detected using an anti-Spy antibody. The oxidized form of Spy (Spy^ox), identified by a mobility shift on SDS-PAGE, performed under reducing conditions, is indicated. Molecular weights are indicated. C. *spy* expression in response to chlorite exposure. β-galactosidase assays were perfomed on WT (LL1322), Δ*cpxR* (LL1324), and Δ*baeR* (LL1378) strains are shown, with error bars representing standard deviations (*n* = 3). The data underlying this

panel can be found in S3 Data. Spy protein levels were also confirmed by western blotting of whole-cell extracts from WT, Δ*cpxR* (LL1198), and Δ*baeR* (LL1538) strains using an anti-Spy antibody. **D.** Spy and MsrP protein levels were analyzed by immunoblotting and detected using anti-Spy and anti-MsrP antibodies. The oxidized form of Spy (Spy$^{ox}$), identified by a mobility shift on SDS-PAGE, is indicated. (B, C, D). Molecular weights are indicated. A nonspecific band is indicated by open circle and was used as an internal loading control stained on the same blot. This blot is representative of at least three independent experiments. **E.** Mass spectrometry analysis of the Spy protein endogenously expressed in a wild-type strain (LL1995) and a Δ*msrP* mutant (LL1996) following induction by chlorite stress. After purification and trypsin digestion, peptides containing at least one methionine residue were analyzed to assess the level of methionine oxidation. The percentage of oxidation for each residue is represented as a heatmap. The mass spectrometry proteomics data for the Spy protein from *Escherichia coli* WT and Δ*msrP* during chlorite stress have been deposited to the ProteomeXchange Consortium via the PRIDE partner repository with the dataset identifier PXD068189 and 10.6019/PXD068189.

our previous findings, the top five proteins with increased abundance included HiuH and MsrP, ranked second and fourth, respectively.

To further investigate Spy production under chlorite stress, we analyzed the protein profile of wild-type *E. coli* cultures exposed to chlorite or left untreated. Immunoblotting revealed a massive accumulation of Spy after chlorite exposure in the wild-type strain, but no signal was detected without chlorite treatment or in a Δ*spy* mutant strain (Fig 1B). Interestingly, we also observed the Spy dimer, even under reducing and denaturing gel conditions (Fig 1B). We further confirmed the induction of *spy* in response to chlorite exposure using a *spy-lacZ* transcriptional fusion that exhibits an ~30-fold increase under chlorite stress conditions (Fig 1C). Together, these results indicate that Spy is among the most overproduced proteins in response to chlorite stress.

## The BaeRS signaling pathway modulates the expression of *spy* in response to chlorite stress

In *E. coli*, *spy* expression is regulated by the CpxRA and BaeRS envelope stress response pathways [11]. To identify the pathway involved during chlorite stress, we quantified *spy-lacZ* expression in different genetic backgrounds (Fig 1C). In Δ*cpxR* cells, *spy* expression increased under chlorite treatment but remained lower than in wild-type cells, whereas no increase was observed in Δ*baeR* cells. (Fig 1C). Additionally, Spy production during chlorite stress was significantly reduced in Δ*baeR* cells compared to wild-type and Δ*cpxR* cells (Fig 1C). These observations indicate that *spy* expression following chlorite treatment is regulated by the BaeRS system rather than by CpxRA (Fig 1C). The role of BaeRS in response to chlorite stress was also confirmed by the global proteomic analysis, which identified several proteins, such as MdtA and MdtB, encoded by Bae-regulated genes among the most abundant under chlorite stress (Fig 1A and S1 Data) [15].

## Spy is specifically induced by chlorite

To determine whether *spy* induction is specific to chlorite or can be triggered by other reactive oxygen species (ROS)/RCS, *spy-lacZ* expression was analyzed following exposure to sublethal concentrations of various oxidants, including $H_2O_2$, paraquat (a superoxide generator), diamide, HOCl, and *N*-chlorotaurine. Tannic acid, a known inducer of *spy* expression [7], was used as a positive control. Among these oxidants, only chlorite significantly increased *spy* expression, while all others caused negligible changes (S1A Fig).

Given that chlorite is produced via chlorate reduction by nitrate reductases under anaerobic conditions [16–18], we investigated whether chlorate induces *spy* expression. Adding 50–200 μM of chlorate to LB medium led to a notable increase in β-galactosidase activity from the *spy-lacZ* fusion (S1B Fig). These results demonstrate that *spy* is also upregulated under anaerobic conditions in the presence of chlorate.

## MsrP repairs chlorite-oxidized Spy

The deletion of *msrP* alters Spy migration during electrophoresis under denaturing conditions (Fig 1B), causing a mobility shift characteristic of Met-O-containing polypeptides [19]. This suggests that (i) chlorite induces both Spy expression

and its oxidation and (ii) that MsrP is involved in the redox control of Spy. To test this hypothesis, we monitored the redox state of Spy in the presence or absence of MsrP. First, in an ΔmsrP strain, expressing msrP from a plasmid restored Spy's reduced mobility, whereas Spy remained oxidized in the strain carrying an empty vector (Fig 1D). MsrP production, induced either by chlorite treatment or plasmid expression, was confirmed by immunoblotting (Fig 1D). Second, we performed a proteomic analysis of Spy-derived peptides containing at least one methionine residue. To minimize artifacts associated with plasmid overexpression, the analyzed proteins were endogenously produced in response to chlorite-induced stress in both a wild-type strain and a ΔmsrP mutant. The results revealed a higher level of Met-O (+16 Da) in peptides derived from Spy produced in the ΔmsrP mutant compared to those from the wild-type strain (Fig 1E and S2 Data). Altogether, these findings indicate that, during chlorite stress, (i) Spy's Met residues are oxidized, and (ii) MsrP repairs and maintains Spy in its reduced form.

## Identification of methionine residues important for Spy function

Whether oxidation of Spy's Met is critical for its chaperone activity is unknown. Spy contains 11 Met residues, some concentrated at the N-terminus in two doublets (Met14-Met15 and Met27-Met28) while others (46, 53,64, 85, 93, 97, and 133) are located more centrally or at the C-terminus part (Fig 2A). Residue numbering is based on the mature form of Spy, excluding the signal peptide sequence. Quan and colleagues [7] identified (Met46, 64, 85, and 97) as conserved among Spy homologs in enterobacteria, proteobacteria, and in some cyanobacteria. Interestingly, three of these Met residues (Met46, Met64, and Met85) are also conserved with the homologous CpxP protein from *E. coli* (Fig 2A) [20,21], and structural analysis revealed that they are located at the core of the substrate-binding cradle (Fig 2B). Based on this analysis, we focused on the N-terminal Met hotspot (Met14-Met15 and Met27-Met28), that we refer to as group 1, and the three core Met (Met46, Met64, and Met85) as group 2. We substituted these Met residues with either Ala (A), to test the physicochemical constraints at each position, or Gln (Q), a Met-O mimetic, to assess the consequences of oxidation at those positions [22,23].

Given that the deletion of *spy* did not result in any strong phenotype under either normal growth or stress conditions, we were unable to assess the functionality of the Met-to-Ala and Met-to-Gln variants through complementation. Instead, we performed a multicopy suppression test, as previously described by the Quan laboratory [24], utilizing the overproduction of Spy to suppress the novobiocin or clindamycin sensitivity in a Δskp ΔfkpA mutant background (Fig 3A). We monitored the antibiotics sensitivity of the Δskp ΔfkpA strain in which the *spy* gene was deleted to prevent expression of the endogenous wild-type *spy,* resulting in the Δskp ΔfkpA Δspy triple mutant. This strain carried plasmids expressing the different Spy variants. Our results show that the group 1 N-terminal Met residues are not essential for Spy's function, as variants with up to four Met-to-Ala or Met-to-Gln substitutions suppressed antibiotic sensitivity as effectively as wild-type Spy (Fig 3B).

The same assay was performed for the group 2 Met (Fig 3C). The Met-to-Ala substitution mutants showed a reduced ability to suppress antibiotic sensitivity, with a stronger effect in the triple mutant (M46A/M64A/M85A) (Fig 3C). Thus, positions 46, 64, and 85 are functionally important, with a low tolerance for substitutions at these positions.

The Met-to-Gln mutants, mimicking Met oxidation, were unable to suppress antibiotic sensitivity (Fig 3C). The double M46Q/M64Q and the triple M46Q/M64Q/M85Q substitutions were particularly affected (Fig 3C). The Spy variant proteins appear to exhibit comparable production levels as indicated by immunoblotting analysis (Fig 3D), with broadly similar banding profiles observed across lanes (S2 Fig).

To gain a broader understanding of the impact of mimicking Met oxidation of other residues—some of which are located on the concave surface of Spy—Met-to-Gln substitution mutants were generated at positions Met53, Met93, Met97, and Met133. Our results show that the M53Q and M133Q single substitutions retain functionality, whereas the M93Q/M97Q double substitutions exhibits impaired function (S3A Fig). Immunoblotting analysis suggests that the

**A**

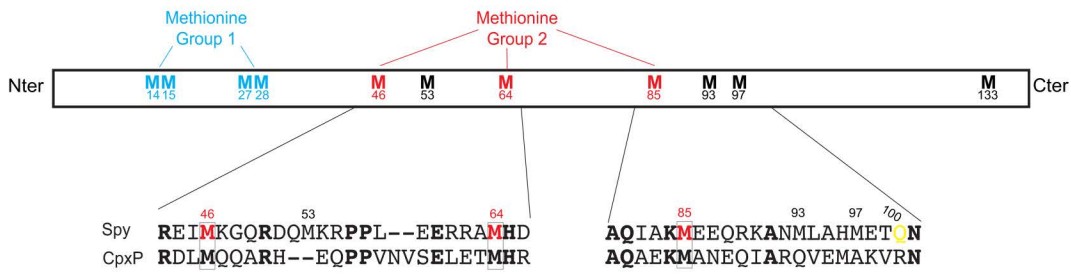

**B**

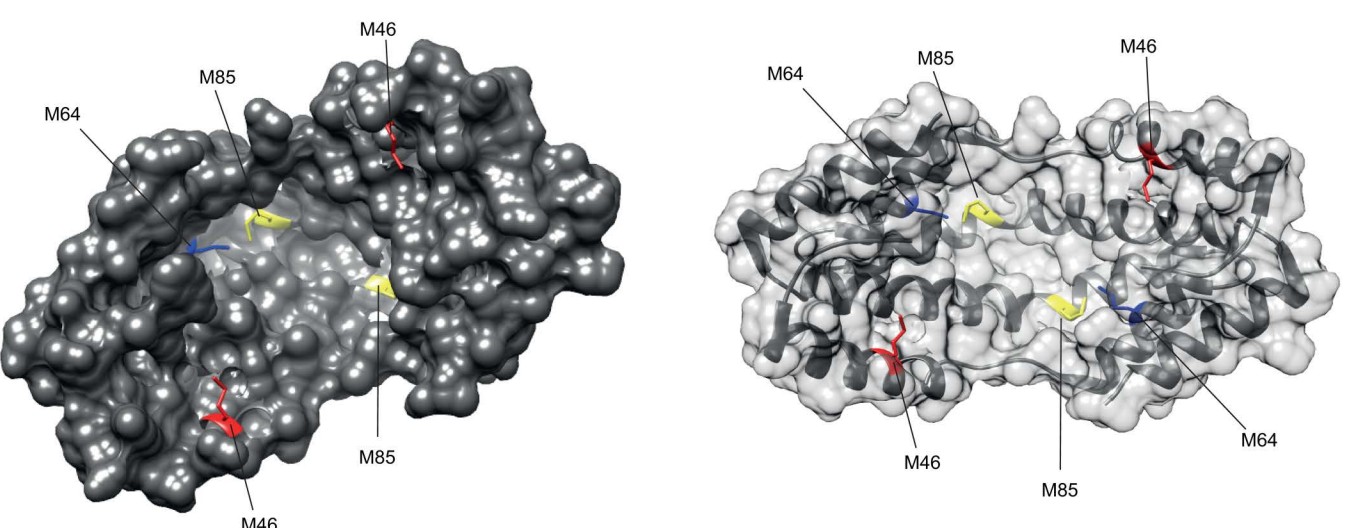

**Fig 2. Structural features of the Spy protein. A.** Schematic of the Spy protein indicating the N-terminal Met hotspot (Met14-Met15 and Met27-Met28, Group 1, in blue) and the core Met residues (Met46, Met64, and Met85, Group 2, in red). Conservation of these Met residues with corresponding residues in the CpxP protein is also shown. **B.** Structure of the *Escherichia coli* Spy protein (PDB:3O39). The Spy dimer is shown in surface view, with core Met residues color-coded: Met46 in red, Met64 in blue, and Met85 in yellow.

Spy variant proteins are produced at comparable levels, and the overall banding patterns appear largely consistent across lanes (S3B Fig). Thus, positions 93 and 97 are functionally important and mimicking their oxidation causes loss of activity.

To rule out potential effects associated with the use of plasmids, we introduced the triple Met-to-Ala (M46A/M64A/M85A) and Met-to-Gln (M46Q/M64Q/M85Q) substitution alleles into the chromosome of a strain lacking *skp* and *fkpA* and carrying the R416C mutation in the *baeS* gene [7], which renders the Bae pathway constitutively active. The resulting Met-to-Gln strain exhibited sensitivity to clindamycin, whereas the Met-to-Ala strain appeared resistant, similar to the strain containing the wild-type *spy* allele (S4 Fig).

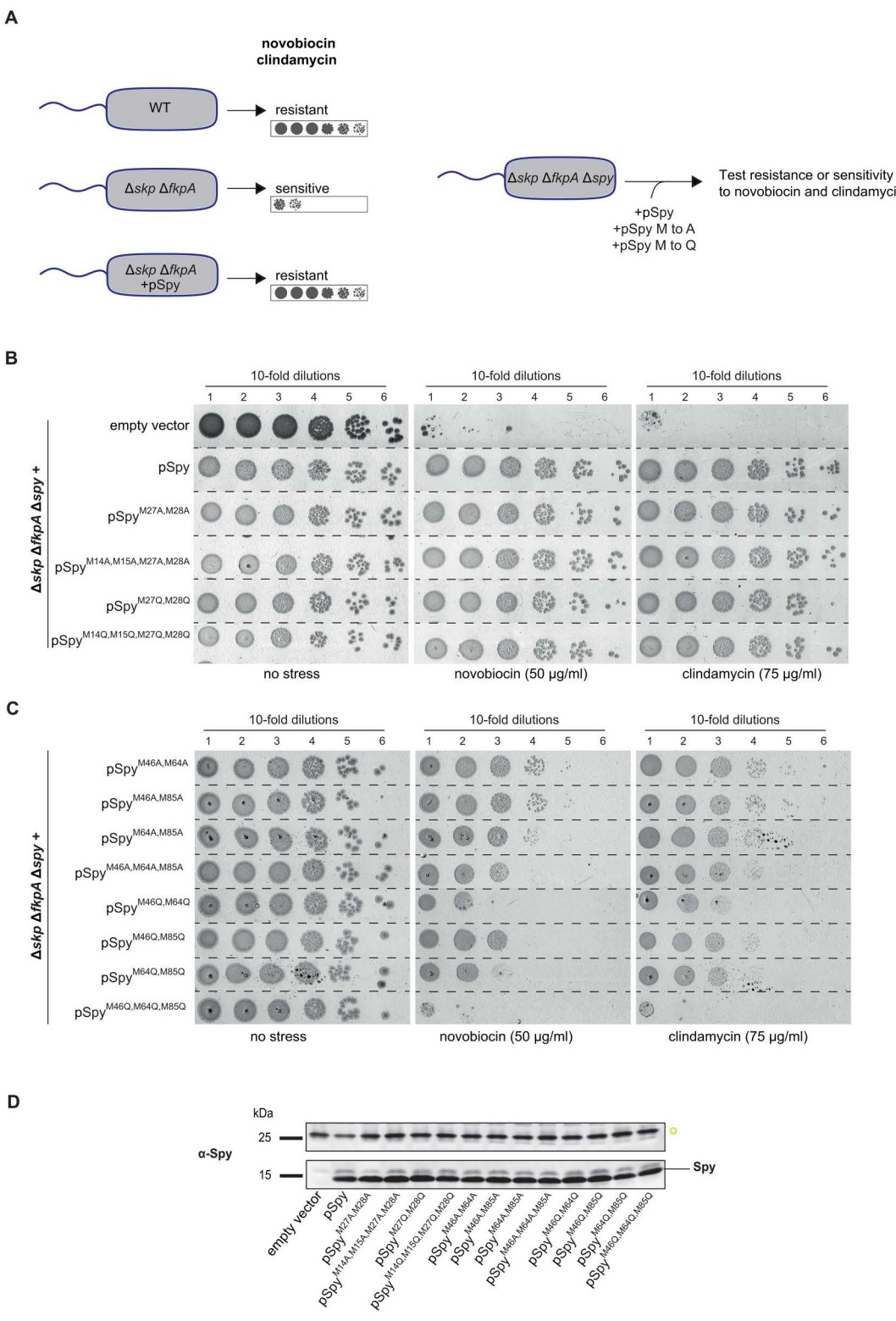

**Fig 3. Importance of Spy's Met residues. A.** Unlike WT *Escherichia coli,* the Δ*skp* Δ*fkpA* strain exhibits sensitivity to novobiocin and clindamycin. Overproduction of Spy suppresses this phenotype, providing a functional assay to evaluate Spy variants. **B** and **C.** Δ*skp* Δ*fkpA* Δ*spy* (LL1414) cells carrying an empty vector or a plasmid expressing wild-type Spy or its variants of group 1 **(B)** or group 2 **(C)** are spotted onto LB plates containing ampicillin

and IPTG, with or without the addition of novobiocin (50 μg/ml) or clindamycin (75 μg/ml). These spotting experiments are representative of three independent experiments. **D.** Evaluation of the production of different Spy variants. LL1414 cells carrying an empty vector or a plasmid expressing wild-type Spy or its variants were cultured overnight at 37 °C onto LB-agar plates supplemented with ampicillin and IPTG. Cells were then resuspended, and protein extracts were prepared for immunoblot analysis. A nonspecific band, marked with an open yellow circle, was used as an internal loading control, and the full membrane scan is provided in (S2 Fig) for improved comparison. This blot is representative of at least three independent experiments.

These genetic screens identified the N-terminal Met14, Met15, Met27, and Met28 as nonessential for Spy's function. In contrast, Met46, Met64, Met85, Met93, and Met97 are crucial and mimicking their oxidation causes loss of activity.

### Methionine oxidation in the cradle impairs Spy's chaperone activity

To determine whether Met oxidation in the cradle region impairs Spy's chaperone activity, we conducted in vivo and in vitro assays.

The chaperone activity of Spy can be evaluated in vivo by monitoring its ability to stabilize the Im7 protein, a known Spy substrate [7]. This is achieved using a stability biosensor, which incorporates the unstable protein Im7-L53A/I54A between the domains of β-lactamase. If Spy stabilizes Im7, the β-lactamase is reconstituted, rendering the strain resistant to penicillin V (Fig 4A) [7,25]. As expected, strains expressing the "Super Spy$^{Q100L}$ Variant" exhibited enhanced resistance to penicillin V compared to cells expressing wild-type Spy (Fig 4B) [25]. In contrast, strains lacking Spy showed sensitivity to the antibiotic (Fig 4B). Remarkably, while the expression of the Met-to-Ala substitution mutant (Spy$^{M46A/M64A/M85A}$) displayed an intermediate antibiotic sensitivity phenotype, we found that expressing the Met-to-Gln substitution mutant (Spy$^{M46Q/M64Q/M85Q}$) phenocopied the absence of Spy (Fig 4B). The Spy variant proteins appear to be produced at comparable levels, with a slight decrease observed for the Spy$^{Q100L}$ variant, as indicated by immunoblotting analysis (Fig 4B); overall banding patterns were broadly similar across lanes (S5 Fig).

To assess the chaperone activity of Spy in vitro, we purified Spy and its variant version and determined their ability to prevent the aggregation of dithiothreitol-reduced α-lactalbumin (α-LA) (Fig 4C) [10,26]. We found that Spy$^{WT}$ prevents α-LA aggregation, whereas the Spy Spy$^{M46Q/M64Q/M85Q}$ variant displayed no anti-aggregation capability (Fig 4C), indicating a loss of chaperone activity. Overall, these results demonstrate that the Spy Spy$^{M46Q/M64Q/M85Q}$ variant is nonfunctional and that mimicking oxidation of Met 46, 64, and 85 results in a loss of Spy's chaperone activity.

To obtain more direct evidence of the impact of Spy oxidation on its activity, Spy was oxidized in vitro using a well-established methodology with $H_2O_2$ as the oxidant [12], then analyzed by mass spectrometry and tested for its functional activity (Fig 4C and 4D). As expected, our results show an increased level of Met-O (+16 Da) following oxidation in vitro treatment. However, the absence of certain peptides in the mass spectrometry data prevents a comprehensive assessment of the oxidation percentage at all methionine positions in Spy. The oxidized form of Spy (Spy-ox) exhibited reduced anti-aggregation activity (Fig 4C); however, this reduction was less pronounced than that observed with the oxidation-mimicking variant. Interestingly, treatment of Spy-ox with MsrP and its complete electron donor system, involving benzyl viologen and dithionite [12], reduced the level of methionine oxidation and restored Spy's activity (Fig 4C and 4D). Altogether, these findings indicate that MsrP ensures Spy's functionality by reducing its Met-O.

### Discussion

In this study, we uncovered that the periplasmic molecular chaperone Spy plays a pivotal role in responding to chlorite stress, a response triggered by the activation of the BaeSR stress pathway. We provided evidence that Spy is vulnerable to oxidation, particularly at three conserved Met residues—Met46, Met64, and Met85—which are critical for its chaperone activity. We further demonstrated that the MsrPQ repair system ensures Spy's functionality by reducing these oxidized residues, safeguarding its role in protein homeostasis during oxidative stress. As such, MsrPQ appears to be an essential player in periplasmic proteostasis. Beyond its role in repairing oxidized Met residues, MsrPQ ensures the functionality of critical molecular chaperones like Spy

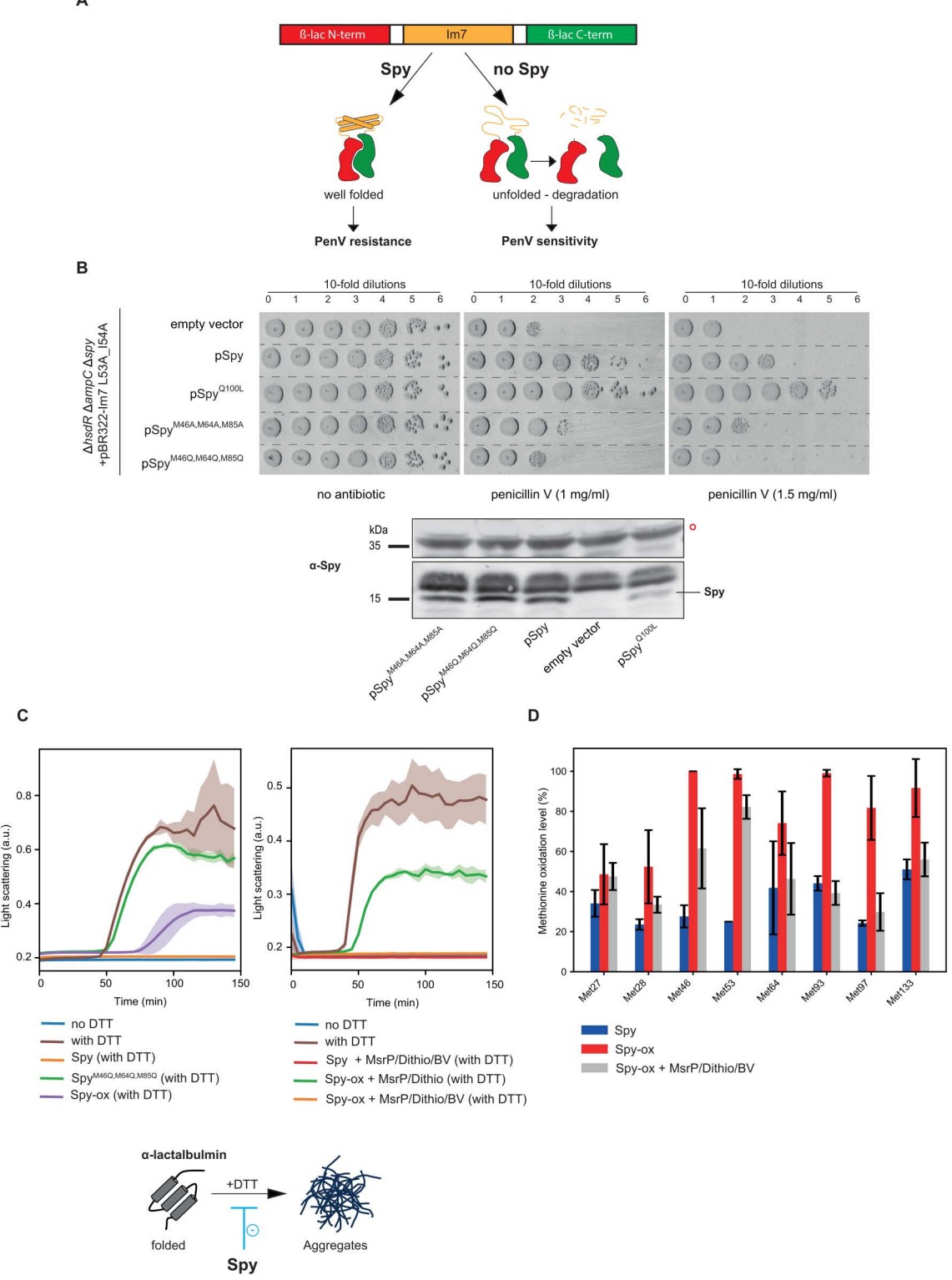

**Fig 4. Impact of Met oxidation on Spy's chaperone activity. A.** The unstable Im7-L53A/I54A protein, a known substrate of Spy, has been inserted between the domains of β-lactamase. In the presence of Spy, Im7 is properly folded, and β-lactamase is reconstituted, making the strain resistant to penicillin V. In the absence of Spy, Im7 is unfolded and degraded, rendering the strain sensitive to penicillin V. **B.** (top) Δ*hsdR* Δ*ampC* Δ*spy* (HW38) cells

carrying plasmids expressing the unstable Im7-L53A/I54A protein and Spy$^{WT}$, Spy$^{Q100L}$, Spy$^{M46A,M64A,M85A}$, or Spy$^{M46Q,M64Q,M85Q}$ were grown in LB supplemented with kanamycin, tetracycline, and IPTG at 37 °C until an OD$_{600}$ of 0.6. 10-fold serial dilutions were spotted onto LB plates containing IPTG, with or without penicillin V (1 and 1.5 mg/ml). These spotting experiments are representative of three independent experiments. (bottom) Evaluation of the production of different Spy variants was evaluated by immunoblot analysis. A nonspecific band, indicated by an open red circle, served as an internal loading control, and the full membrane scan is provided in (S5 Fig) for improved comparison. This blot is representative of at least three independent experiments. **C.** α-lactalbumin aggregation monitored by light scattering over time (left and right) without DTT (blue), with DTT (brown), (left) with DTT, and Spy$^{WT}$ (orange), with DTT and Spy$^{M46Q,M64Q,M85Q}$ (green) and with DTT and Spy-ox (purple); (right) with DTT and Spy$^{WT}$ + MsrP + complete electron donor system (red), with DTT and Spy-ox + MsrP + incomplete electron donor system (green), with DTT and Spy-ox + MsrP + complete electron donor system (orange). The mean values are shown, with standard deviation represented by a shaded area (n = 3). The data underlying this panel can be found in S3 Data. **D.** Spy was treated with H$_2$O$_2$ to induce methionine sulfoxide (Met-O) formation. Subsequently, oxidized Spy was incubated with MsrP in the presence of complete electron donor system. The relative percentage of Met-O in various forms of Spy (native, oxidized [Spy-ox], and repaired [Spy-ox + MsrP/Dithio/BV]) was determined by mass spectrometry analysis (n = 3). Met-O residues were detected in the untreated and MsrP-treated samples owing to limitations inherent to the methodology applied and oxidation of the samples during analytical handling. The data underlying this panel can be found in S3 Data and have also been deposited to the ProteomeXchange Consortium via the PRIDE partner repository with the dataset identifier PXD068073 and 10.6019/PXD068073.

and SurA [12], which collectively prevent protein aggregation and refold damaged proteins [5]. This suggests that MsrPQ likely collaborates closely with molecular chaperones to effectively manage oxidative damage in the periplasm.

Our findings revealed an intriguing paradox: Spy, a molecular chaperone required to mitigate oxidative stress, is itself susceptible to oxidative damage. We demonstrated that mimicking Met oxidation in the disordered N-terminal region has no effect on Spy's activity. This observation is consistent with the role of the N-terminal region in facilitating the release of client proteins without directly contributing to its chaperone activity [10]. We suspect that the N-terminal Met residues may act as sacrificial "oxidant sponges," shielding the chaperone's functional core. This protective role corroborates Stadtman's theory [27], which states that surface-exposed Met residues shield catalytic sites from oxidative damage. Another possibility, not mutually exclusive with the previous one, is that these nonconserved methionine residues are simply not essential for Spy function. In contrast, the oxidation of Met residues in the protein's core renders Spy inactive, highlighting their essential role in maintaining its function. The flexibility of Met side chains, due to the presence of sulfur atoms, facilitates protein–protein interactions. These three Met residues likely play a role for structural adaptation during Spy's interactions with its client proteins. Notably, these Met residues are conserved across all super-variants of Spy obtained from genetic screens aimed at improving chaperone activity against a variety of substrates [25].

Chlorate and chlorite can form naturally in the atmosphere through chlorine photochemistry [28]. However, the primary sources of these compounds are anthropogenic, mainly from the production of bleaching agents and herbicides, leading to environmental contamination. As a result, traces of chlorate and chlorite are found in water and food [29], and could therefore come into contact with the gut microbiota. Our comprehensive proteomic analysis highlights the critical role of Spy and MsrPQ in the bacterial adaptive response to chlorate/chlorite stress. Moreover, it paves the way for future investigations aimed at developing an integrated understanding of this response.

Finally, this work revealed a connection between the oxidative stress response and the envelope stress response in *E. coli*. Indeed, our findings showed that chlorite stress activates not only the HprSR pathway [14], directly linked to oxidative damage, and the BaeSR pathway, traditionally associated with envelope stress [15]. This dual activation suggests an intriguing overlap between these two systems, a link that has received little attention to date. Recent studies in *Salmonella* have similarly observed activation of the Cpx envelope stress pathway by RCS such as *N*-chlorotaurine [30], reinforcing the idea of a broader relationship between oxidative stress and envelope integrity.

## Materials and methods

### Construction of strains and plasmids

All strains were derived from *E. coli* MG1655. Lists of strains, plasmids, and primers used in this study are shown in Tables 1–3, respectively. Gene deletions were performed using the one-step λ Red recombinase chromosomal inactivation system

**Table 1. Strains used in this study.**

| Strain | Genotype and description | Source |
|---|---|---|
| MG1655 | *Escherichia coli* K12 MG1655 (WT) | Laboratory collection |
| CH380 | MG1655 Δ*msrP* | [14] |
| LL1392 | MG1655 Δ*spy* | This study |
| LL1198 | MG1655 Δ*cpxR* | [31] |
| LL1538 | MG1655 *baeR::kan*$^R$ | This study |
| LL1322 | MG1655 *spy-lacZ* | This study |
| LL1324 | MG1655 Δ*cpxR spy-lacZ* | This study |
| LL1378 | MG1655 Δ*baeR spy-lacZ* | This study |
| LL1334 | MG1655 Δ*skp* Δ*fkpA* | This study |
| LL1414 | MG1655 Δ*skp* Δ*fkpA* Δ*spy* | This study |
| LL1590 | MG1655 Δ*skp* Δ*fkpA baeS*$^{R416S}$ | This study |
| LL1596 | MG1655 Δ*skp* Δ*fkpA baeS*$^{R416S}$ *spy*$^{M46A,M64A,M85A}$ | This study |
| LL1594 | MG1655 Δ*skp* Δ*fkpA baeS*$^{R416S}$ *spy*$^{M46Q,M64Q,M85Q}$ | This study |
| HW38 | MG1655 Δ*hsdR* Δ*ampC* Δ*spy* | Shu Quan laboratory |
| LL224 | MG1655 Δ*lacZ* (CmR) | Laboratory collection |
| LL1290 | *msrP-lacZ-ΔzinT* (KanR) | [29] |
| LL1995 | MG1655 *spy-Strep* (KanR) | This study |
| LL1996 | MG1655 Δ*msrP spy-Strep* (KanR) | This study |
| BW25113 | Δ(*araD-araB*)567 Δ(*rhaD-rhaB*)568 Δ*lacZ*4787 (::*rrnB-3*) *hsdR*514 *rph-1* | [32] |

[31]. Deletions were transferred to the WT strain using P1 transduction procedures and verified by polymerase chain reaction (PCR). The antibiotic resistance cassette was removed using Flp recombinases from plasmid pCP20 [31].

**Strain engineering.** The *spy::lacZ* fusion strain (LL1322) was constructed at the *spy* gene locus. A PCR product containing the *spy* promoter fused to *lacZ* was amplified from strain LL1290 and recombined into BW25113 carrying pKD46. The resulting construct was subsequently transduced into strain LL224 (MG1655 Δ*lacZ*::Cm$^R$). The chromosomal *baeS-R416C* strain (LL1590) was generated following procedure described in [32]. The *baeS-R416C sce-I::cat* fragment was amplified by PCR (primers LL533/LL534, pWRG100 template) and electroporated into MG1655 cells expressing λ recombinase from pWRG99. After selection on chloramphenicol (25 μg/mL), the construct was transduced into recipient strains (MG1655, LL1334), and the Cm cassette was removed via λ Red recombination using primers LL535/LL536. The *baeS-R416C* mutation was integrated by electroporation, selected on ampicillin (100 μg/mL) and aTc (1 μg/mL), and confirmed by sequencing. Strains carrying chromosomal *spy* mutant alleles were constructed as follows: plasmids pAV263 and pAV264, containing the mutated versions of *spy*, were opened at the *Hind*III sites. A kanamycin resistance cassette, extracted by *Hind*III digestion from pKD4 vector, was inserted and ligated. These constructs were used as templates for PCR amplification with primers LL511 and LL321 to perform chromosomal insertion by λ Red recombination [33]. The chromosomal *spy-Strep-KanR* strain (LL1995) was generated by electroporating a PCR product obtained using primers LL717 and LL321, with chromosomal *spy-KanR* DNA as the template.

**Plasmid engineering.** The IPTG-inducible expression vector for *spy* was constructed by amplifying the *spy* gene from the MG1655 chromosome using primers LL318 and LL319. The PCR product was cloned into pTrc99A via *EcoR*I and *Sal*I restriction sites, generating plasmid pLL212. Plasmids pLL333 and pLL334 (pCDF-Spy$^{M46A,M64A,M85A}$ and pCDF-Spy$^{M46Q,M64Q,M85Q}$, respectively) were constructed by amplifying mutated *spy* alleles from strains LL1596 and LL1594 using primers LL668 and LL669. The PCR products were digested with *BamH*I and cloned into the *BamH*I-linearized pCDFtrcKan vector. For Spy purification, plasmid pLL323 (pJF-no_ss-spy-StrepTag) was constructed by amplifying the *spy* gene (without its signal sequence) from MG1655 using primers LL657 and LL628. This included a Strep-tag II coding sequence at the 3′ end. The PCR product was cloned into

**Table 2. Plasmids used in this study.**

| Plasmid | Genotype and description | Source |
|---|---|---|
| pAG192 | pTAC-MAT-Tag-2NdeI MsrP, AmpR (pMsrP) | [12] |
| pLL212 | pTrc99A-spy (pSpy) | This study |
| pLL219 | pTrc99A-spyM27Q,M28Q (pSpy$^{M27Q,M28Q}$) | This study |
| pLL220 | pTrc99A-spyM27A,M28A (pSpy$^{M27A,M28A}$) | This study |
| pAV221 | pTrc99A-spyM14Q,M15QM27Q,M28Q (pSpy$^{M14Q,M15Q,27Q,M28Q}$) | This study |
| pAV222 | pTrc99A-spyM14A,M15A,M27A,M28A (pSpy$^{M14A,M15A,27A,M28A}$) | This study |
| pAV239 | pTrc99A-spyM46Q (pSpy$^{M46Q}$) | This study |
| pAV240 | pTrc99A-spyM64Q (pSpy$^{M64Q}$) | This study |
| pAV241 | pTrc99A-spyM85Q (pSpy$^{M85Q}$) | This study |
| pAV242 | pTrc99A-spyM46A (pSpy$^{M46A}$) | This study |
| pAV243 | pTrc99A-spyM64A (pSpy$^{M64A}$) | This study |
| pAV2414 | pTrc99A-spyM85A (pSpy$^{M85A}$) | This study |
| pAV251 | pTrc99A-spyM46Q,M64Q (pSpy$^{M46Q,M64Q}$) | This study |
| pAV252 | pTrc99A-spyM46Q,M85Q (pSpy$^{M46Q,M85Q}$) | This study |
| pAV253 | pTrc99A-spyM64Q,M85Q (pSpy$^{M64Q,M85Q}$) | This study |
| pAV254 | pTrc99A-spyM46A,M64A (pSpy$^{M46A,M64A}$) | This study |
| pAV255 | pTrc99A-spyM46A,M85A (pSpy$^{M46A,M85A}$) | This study |
| pAV256 | pTrc99A-spyM64A,M85A (pSpy$^{M64A,M85A}$) | This study |
| pAV263 | pTrc99A-spyM46Q,M64Q,M85Q (pSpy$^{M46Q,M64Q,M85Q}$) | This study |
| pAV264 | pTrc99A-spyM46A,M64A,M85A (pSpy$^{M46A,M64A,M85A}$) | This study |
| pSpy | pCDFTrckanBamHI Spy WT (KanR) | Shu Quan laboratory |
| pSpyQ100L | pCDFtrcKanBamH1Spy Q100L (KanR) | Shu Quan laboratory |
| pIm7-L53AI54A | pBR322bla::GSlinker Im7 L53AI54A (TetR) | Shu Quan laboratory |
| pLL333 | pCDFtrcKanBamH1SpyM46A,M64A,M85A (KanR) | This study |
| pLL334 | pCDFtrcKanBamH1SpyM46Q,M64Q,M85Q (KanR) | This study |
| pLL323 | pJF119-no_ss_spy strep-tag (AmpR) | This study |
| pLL326 | pJF119- no_ss_spy M46Q,M64Q,M85Q strep-tag (AmpR) | This study |

pJF119EH using *Eco*RI and *Hind*III restriction sites. The same procedure was used to construct the plasmid pLL326 for expressing Spy$^{M46Q,M64Q,M85Q}$-strepTag, using LL1594 chromosome as the template. All constructs were verified by sequencing.

## Bacterial cultures

*E. coli* strains were streaked onto an LB-agar plate and incubated overnight at 37 °C. The following day, a single colony was inoculated into LB and cultured aerobically with shaking at 37 °C overnight. The next morning, cultures were diluted to 1:100 prior to performing the experiments described in the following. For immunoblot analyses and β-galactosidase assays, cultures were split into two subcultures and 400 μM $ClO_2^-$ was added to one of the cultures before an additional 4 hours incubation at 37 °C with shaking. The following antibiotic concentrations were used for plasmid maintenance: 100 μg mL$^{-1}$ ampicillin, 25 μg mL$^{-1}$ chloramphenicol, 30 μg mL$^{-1}$ kanamycin, 25 μg mL$^{-1}$ tetracycline.

## Sample preparation for proteomic analysis

200 μL of the overnight WT culture was diluted into 20 mL of fresh LB medium in a 50 mL Falcon tube. This dilution was split into two 10 mL aliquots in separate 250 mL glass flasks and incubated at 37 °C with shaking for 2 hours. A final concentration of 400 μM $ClO_2^-$ was added to one of the flasks. The cultures were incubated at 37 °C with shaking for

**Table 3. Primers used in this study.**

| Name | Sequence (5′–3′) | Used for |
|------|------------------|----------|
| AV387 | aggcaagttcggtccgcatcaggacCAGCAGttcaaag | pLL219 |
| AV388 | tgcggaccgaacttgcctttgtggtgCTGCTGcg | pLL219 |
| AV389 | aggcaagttcggtccgcatcaggacGCGGCAttcaaag | pLL220 |
| AV390 | tgcggaccgaacttgcctttgtggtgTGCCGCcg | pLL220 |
| AV409 | gaagccgCAGCAGcaccacaaaggcaagttcggtcc | pLL221 |
| AV410 | gtggtgCTGCTGcggcttcgcgtcagccggtgc | pLL221 |
| AV411 | gaagccgGCGGCAcaccacaaaggcaagttcggtcc | pLL222 |
| AV412 | gtggtgTGCCGCcggcttcgcgtcagccggtgc | pLL222 |
| AV473 | cgaaatcCAGaaaggccagcgtgaccagatgaaacg | pAV239 pAV251 pAV252 |
| AV474 | gcctttCTGgatttcgcggatctgctgtttctgc | pAV239 pAV251 pAV252 |
| AV475 | cgcgcaCAGcatgacatcattgccagcgataccttcgataaag | pAV240 |
| AV476 | tgtcatgCTGtgcgcggcgttcttccagc | pAV240 |
| AV477 | tcgcaaaaCAGgaagaacagcgcaaagctaacatgc | pAV241 pAV253 pAV263 |
| AV478 | gttcttcCTGttttgcgatctgcgcttcagcttttactttatcg | pAV241 pAV253 pAV263 |
| AV479 | cgaaatcGCAaaaggccagcgtgaccagatgaaacg | pAV242 pAV254 pAV255 |
| AV480 | gcctttTGCgatttcgcggatctgctgtttctgc | pAV242 pAV254 pAV255 |
| AV481 | cgcgcaGCAcatgacatcattgccagcgataccttcgataaag | pAV243 |
| AV482 | tgtcatgTGCtgcgcggcgttcttccagc | pAV243 |
| AV483 | tcgcaaaaGCAgaagaacagcgcaaagctaacatgc | pAV244 pAV256 pAV264 |
| AV484 | gttcttcTGCttttgcgatctgcgcttcagcttttactttatcg | pAV244 pAV256 pAV264 |
| LL657 | ccgGAATTCatggcagacaccactaccgcagcaccggctgacg | pLL323 |
| LL628 | ttaagcttttactttttcgaactgcgggtggctccattcagcagttgcaggcattttaccttttgccg | pLL323 |
| LL533 | tatcgcaccgaaggttccTgcaaccgtgccagcggcggttccGTGTAGGCTGGAGCTGCTTC | LL1590 |
| LL534 | cccgccaaaaggcgaatgggcagcaataatgcgcctATTACCCTGTTATCCCTAcct-tagttcctattccgaagttc | LL1590 |
| LL535 | cgcaccgaaggttccTgcaaccgtgccagc | LL1590 |
| LL536 | cccgccaaaaggcgaatgggcagcaataat | LL1590 |
| LL318 | GCTGAATTCATGcgtaaattaactgcactgtttgttgc | pLL212 |
| LL319 | CTCGCTCGAGttattcagcagttgcagg | pLL212 |
| LL511 | gaaagccgtaataaataactgaaaggaaggatatagaatAtgcgtaaattaactgcactgtttgttgc | LL1596 LL1594 |
| LL321 | tggacaagaccggcggtcttaagttttttggctgaaagattaCATATGAATATCCTCCTTAG | LL1596 LL1594 |

**Table 3.** (Continued)

| Name | Sequence (5′–3′) | Used for |
|------|------------------|----------|
| LL355 | gaaagccgtaataaataactgaaaggaaggatatagaatatggtcgtttacaacgtcgtgactgg-gaaaaccct | LL1322 |
| LL357 | tggacaagaccggcggtcttaagtttttggctgaaagaacttactaaagcggcatcgaggcgttatcat-gagaatac | LL1322 |
| LL668 | Cagggatccatgcgtaaattaactgcactgtttgttgc | pLL333<br>pLL334 |
| LL669 | atatgggatccttattcagcagttgcag | pLL333<br>pLL334 |
| LL717 | tgcctgcaactgctgaatggagccacccgcagttcgaaaagtaaGTGTAGGCTGGAGCT-GCTTC | LL1995 |

an additional 4 hours. 1 mL of each culture was transferred to pre-weighed 1.5 mL Eppendorf tubes and centrifuged at 8,000 rpm for 2 min. The supernatants were removed and the tubes containing the bacterial pellets were weighed again to determine the pellet mass. For every 10 mg of bacterial pellet, 60 μL of 2× Tris-Glycine SDS buffer without reducing agents (Novex, LC2676) was added. The tubes were heated at 99 °C for 5 min.

## Shotgun label-free proteomics

Protein extracts (40 μg of total proteins) from three biological replicates under two conditions (cells grown in LB medium, unexposed and exposed to $ClO_2^-$) were subjected to NuPAGE electrophoresis for a short migration of 4 min (Fig 1A). The proteomes were treated and subjected to trypsin proteolysis as previously described [34]. For (Fig 1E), the polyacrylamide bands corresponding to the Spy protein were treated and subjected to trypsin proteolysis as previously described [34]. The resulting peptides were analyzed on an Orbitrap Exploris 480 (Thermo Scientific) tandem mass spectrometer coupled to a Vanquish Neo UHPLC (Thermo Scientific) and operated as reported [35]. Peptides were desalted on a reverse-phase PepMap Neo C18 μ-precolumn (100 Å, 300 μm i.d. × 5 mm, Thermo Scientific) and separated on a EasySpray PepMap Neo C18 column (2, 75 μm i.d. × 500 mm, Thermo Scientific) at a flow rate of 250 nL/min using a 60-min gradient (5%–25% B from 0 to 60 min) followed by 5-min gradient (25%–40% B) from 60 to 65 min) of mobile phase A (0.1% HCOOH/99.9% $H_2O$) and phase B (0.1% HCOOH/99.9% $CH_3CN$). The mass spectrometer was operated in data-dependent acquisition (DDA) mode with a full mass scan from $m/z$ 350–1,500, a precursor resolution of 120,000 and a MS/MS resolution of 15,000. The top 20 precursor ions detected at each scan cycle, with potential charge state 2+ or 3+, were sequentially selected and subjected to fragmentation, using a dynamic exclusion time of 10 s.

In Fig 4D, the nine purified Spy samples were subjected to Trypsin Gold (Promega) proteolysis using the SP3 procedure, as previously described [36]. The resulting peptides were purified with SPE, eluted, and acidified with 0.1% trifluoroacetic acid. A quantity of 20 ng of peptides was analyzed on an Orbitrap Astral (Thermo Scientific) tandem mass spectrometer coupled to a Vanquish Neo UHPLC (Thermo Scientific) operated as reported [35]. Peptides were desalted on a reverse-phase PepMap Neo Trap precolumn (100 Å, 300 mm i.d. × 5 mm, Thermo Scientific) and resolved on a AuroraXT column (75 μm i.d. × 25 cm, Ionopticks) at a flow rate of 400 nL/min using a 30-min gradient (3%–25% B from 0 to 30 min) followed by a 5 min gradient (25%–36% B from 30 to 35 min) of mobile phase A (0.1% HCOOH/99.9% $H_2O$) and phase B (0.1% HCOOH/99.9% $CH_3CN$). The Orbitrap Astral mass spectrometer was operated in data-dependent acquisition (DDA) mode with a full mass scan from $m/z$ 350–1,500, a precursor resolution of 120,000 (at $m/z$ 100) and a MS/MS resolution of 80,000 (at $m/z$ 524) with cycle time set at 1 sec. Precursors with potential charge state 2+ or 3+ were selected for fragmentation with a dynamic exclusion time of 10 s. Tandem mass spectrometry results were interpreted with the Mascot Daemon 2.6.1 search engine (Matrix Science) as recommended against the *E. coli* annotated proteome (4,531

entries) [33]. The mass spectrometry proteomics data have been deposited to the ProteomeXchange Consortium via the PRIDE partner repository with the dataset identifier PXD068073, PXD068145, and PXD068189.

## Immunoblot analysis

Similar cell densities (10 units/mL; normalized by $A_{600}$) were pelleted by centrifugation (3,000$g$, 5 min), resuspended in Laemmli SDS buffer with reducing agents (containing DTT (50 mM) and β-mercaptoethanol (5%)), heated for 10 min at 95 °C, and separated by gel electrophoresis (Novex 4%–20% Tris-glycine Gel (ThermoFisher Scientific)). Samples were transferred to 0.2 μm nitrocellulose membranes and probed with anti-Spy (Cliniosciences) and anti-MsrP antibody (provided by the Jean-François Collet Lab, de Duve Institute, UCLouvain), followed by an HRP-conjugated anti-Rabbit (Promega W4011) for Spy and anti-guinea pig IgG secondary antibody (Sigma-Aldrich) for MsrP. Chemiluminescence signal was collected using an ImageQuant Las4000 camera (GE Healthcare). The presented results are representative of at least three independent experiments.

## β-galactosidase assays

100 μL of culture treated or untreated with 400 μM $ClO_2^-$ was added to 900 μL of β-galactosidase buffer. Levels of β-galactosidase were measured as previously described [37]. The same protocol was employed to assess the effect of other oxidants on *spy* expression. The oxidants were added to LB medium to achieve final concentrations of 5 mM $H_2O_2$ (Honeywell), 0.3 mM paraquat (Sigma-Aldrich), 0.3 mM diamide (Sigma-Aldrich), 4 mM HOCl (Honeywell), and 1 mM *N*-chlorotaurine. 0.25 mM tannic acid (Sigma-Aldrich) was used as a control.

## *N*-ChT synthesis

*N*-ChT (Cl-HN-$CH_2$-$CH_2$-$SO_3^-$) was produced by mixing 100 mM HOCl (Honeywell) and 100 mM taurine (Sigma-Aldrich) in 0.1 M phosphate potassium buffer (pH 7.4). After 10 min at room temperature, the *N*-ChT concentration was determined by measuring the absorbance at 252 nm ($\varepsilon = 429$ M$^{-1}$ cm$^{-1}$) and stored at 4 °C.

## Site-directed mutagenesis

50 μL PCR reactions were performed using Q5 Hot start High-Fidelity DNA polymerase (New England Biolabs). The resulting PCR products were digested with *Dpn*I, purified using GeneJET PCR purification kit (Thermo Fisher), and transformed into DH5α. Three colonies were randomly selected from each transformation, and the plasmids were purified using GeneJET Plasmid Miniprep kit (Thermo Fisher) and verified by sequencing. Templates and primers used for generating Spy variants are shown in Table 4.

## Novobiocin and clindamycin survival assays

The Δ*skp* Δ*fkpA* Δ*spy* strain (LL1414) harboring an empty vector, pSpy wild-type or mutated variants grew aerobically in LB supplemented with ampicillin and IPTG (0.1 mM) at 37 °C. At OD$_{600}$ = 1, cultures were serially diluted in phosphate-buffered saline (PBS). 3.5 μL of 10-time serial dilutions were spotted onto LB-agar plates containing ampicillin and IPTG (0.1 mM) supplemented or not with 50 μg/mL novobiocin or 75 μg/mL clindamycin. Plates were incubated at 37 °C overnight. For the clindamycin sensitivity experiments without plasmid addition, the protocol was identical, except ampicillin and IPTG were omitted. LB plates supplemented with 50 μg/mL clindamycin were used.

## Chaperone activity assay in vivo

Spot titer experiments were performed to quantify the relative in vivo chaperone activity of Spy variants, as previously established and described [38]. Briefly, Δ*hsdR* Δ*ampC* Δ*spy* cells (HW38) expressing Spy variants from a

**Table 4. Templates et primers used to construct *spy* variants.**

| Templates | Primers | Products |
|---|---|---|
| pLL212 | AV387/AV388 | pLL219 |
| pLL212 | AV389/AV390 | pLL220 |
| pLL219 | AV409/AV410 | pLL221 |
| pLL220 | AV411/AV412 | pLL222 |
| pLL212 | AV473/AV474 | pAV239 |
| pLL212 | AV475/AV476 | pAV240 |
| pLL212 | AV477/AV478 | pAV241 |
| pLL212 | AV479/AV480 | pAV242 |
| pLL212 | AV481/AV482 | pAV243 |
| pLL212 | AV483/AV484 | pAV244 |
| pAV240 | AV473/AV474 | pAV251 |
| pAV241 | AV473/AV474 | pAV252 |
| pAV240 | AV477/AV478 | pAV253 |
| pAV243 | AV479/AV480 | pAV254 |
| pAV244 | AV479/AV480 | pAV255 |
| pAV243 | AV483/AV484 | pAV256 |
| pAV251 | AV477/AV478 | pAV263 |
| pAV254 | AV483/AV484 | pAV266 |

pCDFTrckanBamHI and βla-Im7 L53A I54A sensor from pBR322 plasmids (provided by the Shu Quan Lab, Shanghai, Jiao Tong University) were grown in LB supplemented with kanamycin, tetracycline and IPTG (0.1 mM) to $OD_{600} = 0.6$, serially diluted, and plated onto LB-agar plates supplemented or not with penicillin V (1 and 1.5 mg/ml) and IPTG (0.1 mM). Plates were incubated at 37 °C overnight.

## Purification of Spy

Strep-tagged versions of wild-type Spy and its variants were expressed and purified from Δ*spy* (LL1392) cells harboring plasmids pLL323 or pLL326, which overexpressed $Spy^{WT}$ or $Spy^{M46Q,M64Q,M85Q}$ proteins, respectively. Cells were grown aerobically at 37 °C in LB medium supplemented with ampicillin. When cultures reached $OD_{600} = 0.6$, protein expression was induced with IPTG (0.1 mM final concentration) for 4 hours at 37 °C. The pellet from a 400 ml culture was resuspended in buffer A (100 mM Tris-HCl, 50 mM NaCl, pH 8). Cells were disrupted by two passes through a French press. After centrifugation at 12,000 rpm for 30 min at 4 °C to remove debris, the supernatant was loaded onto a 5-ml Strep-Trap HP column (GE Healthcare) equilibrated with buffer A. The column was washed with buffer A, and Spy was eluted using buffer A supplemented with 2.5 mM desthiobiotin. Pure fractions were pooled, and desthiobiotin was removed using an Amicon Ultra-10K filter (Millipore).

## Spy oxidation and repair in vitro

Spy protein was oxidized using $H_2O_2$ (50 mM) for 2 hours at 37 °C. The reaction was stopped by buffer exchange using Zeba Spin Desalting Columns (7K MWCO) with buffer A. The resulting oxidized Spy protein (Spy-ox) was then treated with MsrP enzyme in the presence of a reducing system to generate the repaired form. Specifically, 100 µM Spy-ox was incubated for 2 hours at 30 °C with 4 µM purified MsrP, 10 mM benzyl viologen (BV), and 10 mM dithionite (Dithio). The reaction was stopped by buffer exchange using Zeba Spin Desalting Columns (7K MWCO) with buffer A. As a control, Spy-ox was treated under the same conditions, except that the electron donor system lacked BV.

## α-lactalbumin aggregation assay

Bovine α-lactalbumin aggregation assay was performed as previously described [10,25,26]. Briefly, aggregation of α-LA (type III, Sigma-Aldrich) at 100 μM was initiated by adding 20 mM DTT to a buffer containing 50 mM phosphate buffer, 100 mM sodium chloride, and 5 mM EDTA, pH 7.0. The light scattering of α-LA in the absence or presence of Spy (30 μM) was monitored at 360 nm using a Spark microplate reader (TECAN) with a 5-min detection period for 600 min at 25 °C.

## Supporting information

**S1 Fig. Spy induction is specific to chlorite stress.** *S*py expression in response to various oxidants. **A.** β-galactosidase assays of WT (LL1322) treated aerobically with 5 mM $H_2O_2$, 0.3 mM paraquat, 0.3 mM diamide, 4 mM HOCl, 1 mM *N*-ChT, 0.4 mM $NaClO_2$, or 0.25 mM tannic acid for 4 hours. **B.** β-galactosidase assays were performed on WT (LL1322) treated anaerobically overnight with $NaClO_3$ (6.25–200 μM). Error bars indicate standard deviations ($n = 3$). The data underlying this figure can be found in S3 Data.
(EPS)

**S2 Fig. The full membrane scan from which the blot shown in Fig 3D was derived is provided to allow improved comparison of the loading control.**
(EPS)

**S3 Fig.** **A.** Δ*skp* Δ*fkpA* Δ*spy* (LL1414) cells carrying an empty vector or a plasmid expressing wild-type Spy or its variants Spy[M53Q,] Spy[M93Q,M97Q], Spy[M133Q] are spotted onto LB plates containing ampicillin and IPTG, with or without the addition of novobiocin (50 μg/ml) or clindamycin (75 μg/ml). These spotting experiments are representative of three independent experiments. **B.** Evaluation of the production of different Spy variants. LL1414 cells carrying an empty vector or a plasmid expressing wild-type Spy or its variants were cultured overnight at 37 °C onto LB-agar plates supplemented with ampicillin and IPTG. Cells were then resuspended, and protein extracts were prepared for immunoblot analysis. The full membrane scan is provided in for improved comparison, and a nonspecific band, indicated by an open yellow circle, served as an internal loading control. This blot is representative of at least three independent experiments.
(EPS)

**S4 Fig. Substitution alleles into the chromosome.** WT, Δ*skp* Δ*fkpA* (LL1334), Δ*skp* Δ*fkpA baeS*[R416S] (LL1590), Δ*skp* Δ*fkpA baeS*[R416S] *spy*[M46Q,M64Q,M85Q] (LL1594), Δ*skp* Δ*fkpA baeS*[R416S] *spy*[M46A,M64A,M85A] (LL1596) are spotted onto LB plates supplemented with or without clindamycin (50 μg/mL).
(EPS)

**S5 Fig. The full membrane scan from which the blot shown in Fig 4B was derived is provided to allow improved comparison of the loading control.**
(EPS)

**S1 Data. List of proteins and their spectral counts used for Fig 1A.** Protein extracts from three biological replicates under two conditions (cells grown in LB medium LB A1, A2, and A3; and cells grown in LB medium exposed to $ClO_2^-$ LBCLO2 B1, B2, and B3) were subjected to NuPAGE electrophoresis for a short migration. The proteomes were treated and subjected to trypsin proteolysis. The resulting peptides were analyzed on an Orbitrap Exploris 480 tandem mass spectrometer coupled to a Vanquish Neo UHPLC. A protein is validated when at least two peptides were found in one of the samples and at least one of these peptides is unambiguously assigned to this protein (this peptide is then proteotypic of the protein).
(XLSX)

**S2 Data. Number of peptides analyzed used for the proteomic analysis in Fig 1E.** Proteomic analysis of Spy-derived peptides containing at least one methionine residue in both a wild-type strain and a Δ*msrP* mutant following chlorite stress. The total number of each peptide analyzed is indicated in the rightmost column. For each peptide, the number of peptides containing one methionine oxidation (or two, when multiple methionine residues are present) is provided. The number of peptides carrying a deamidation modification is also shown, as well as the number of peptides with no detectable modifications (reduced form).
(XLSX)

**S3 Data. Numerical values.** All numerical values used to generate Figs 1C, 4C, 4D S1A, and S1B are provided in this document. It includes replicate values, means, and errors, presented in the order in which they appear in the manuscript.
(XLSX)

**S1 Raw Images. Uncropped versions of all western blot images presented in the main figures and Supporting information.** The imaging method used to capture the blots is indicated. Lanes marked with an 'X' correspond to those not included in the final figures.
(PDF)

## Acknowledgments

We thank members of the Ezraty group for their discussions. We are grateful to James Bardwell (University of Michigan), Shu Quan (Shanghai Jiao Tong University), and Jean-François Collet (de Duve Institute, UCLouvain) for their advice, discussions, and for providing strains and plasmids. We also thank Marianne Ilbert and Olivier Genest (BIP-AMU/CNRS Marseille) for their valuable advice, as well as Mélodie Kielbasa (CEA-Li2D) for her technical help with mass spectrometry.

## Author contributions

**Conceptualization:** Laurent Loiseau, Alexandra Vergnes, Jean Armengaud, Maxence S. Vincent, Benjamin Ezraty.

**Data curation:** Laurent Loiseau, Nathan De Visch, Jean Armengaud, Maxence S. Vincent, Benjamin Ezraty.

**Formal analysis:** Laurent Loiseau, Alexandra Vergnes, Jean Armengaud, Maxence S. Vincent, Benjamin Ezraty.

**Funding acquisition:** Jean Armengaud, Maxence S. Vincent, Benjamin Ezraty.

**Investigation:** Laurent Loiseau, Alexandra Vergnes, Maxence S. Vincent, Benjamin Ezraty.

**Methodology:** Laurent Loiseau, Nathan De Visch, Alexandra Vergnes, Jean Armengaud, Maxence S. Vincent, Benjamin Ezraty.

**Project administration:** Benjamin Ezraty.

**Resources:** Benjamin Ezraty.

**Supervision:** Benjamin Ezraty.

**Validation:** Maxence S. Vincent, Benjamin Ezraty.

**Writing – original draft:** Jean Armengaud, Maxence S. Vincent, Benjamin Ezraty.

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
