## [Editor Report · Decision Letter 0]

20 Dec 2024

Dear Ben, 

Thank you for submitting your manuscript entitled "Maintaining the methionine residues of the chaperone Spy in a reduced state is crucial for periplasmic proteostasis." for consideration as a Research Article by PLOS Biology.

Your manuscript has now been evaluated by the PLOS Biology editorial staff, as well as by two academic editors with relevant expertise. However, during discussion with the Academic Editors an important concern was raised. Both are concerned about the physiological relevance of the finding, specially since E. coli cells lacking Spy to not seem to have an increased sensitivity with chlorite stress. The other Academic Editor, suggested that given the limited mechanistic insights and that the physiological importance needs further investigation, we could invite you to submit the study as a Short Report. We would of course like more a study showing physiological relevance, but we are open to invite you to submit the study as a Short Report. Please just remember that this would require 4 figures. Given the upcoming holidays, I am extending the deadline of submission to Dec 27. 

If you decide to submit as a Short Report, we need you to complete your submission by providing the metadata that is required for full assessment. To this end, please login to Editorial Manager where you will find the paper in the 'Submissions Needing Revisions' folder on your homepage. Please click 'Revise Submission' from the Action Links and complete all additional questions in the submission questionnaire. Please, when adding the rest of the metadata choose "Short Report".

Once your full submission is complete, your paper will undergo a series of checks in preparation for peer review. After your manuscript has passed the checks it will be sent out for review. To provide the metadata for your submission, please Login to Editorial Manager (https://www.editorialmanager.com/pbiology) within two working days, i.e. by Dec 27 2024 11:59PM.

Kind regards,

Melissa

Melissa Vazquez Hernandez, Ph.D.

Associate Editor

PLOS Biology

---

## [Decision Letter · Decision Letter 1]

19 Feb 2025

Dear Ben,

Thank you for your patience while your manuscript "Maintaining the methionine residues of the chaperone Spy in a reduced state is crucial for periplasmic proteostasis." was peer-reviewed at PLOS Biology. It has now been evaluated by the PLOS Biology editors, an Academic Editor with relevant expertise, and by four independent reviewers. 

In light of the reviews, which you will find at the end of this email, we would like to invite you to revise the work to thoroughly address the reviewers' reports. 

As you will see below, the majority of reviewers are positive about the relevance and novelty of the study, yet some concerns have raised during revision. Reviewer 1 is not convinced that Spy is a target of chlorite stress. The reviewer recommends a different approach to see if methionine oxidation does affect the activity of Spy and mass spec characterization to show which residues are actually oxidized. This reviewer also requests to evaluate if MsrP would also recover the activity of Spy in vitro. Reviewer 2 asks that you clarifiy the number of replicates in ceratain experiments. Reviewer 3 requests that you confirm the oxidation through mass-spec, if MsrPQ can recover Spy activity in vitro, and if Spy aggregates when oxidized. This reviewer also would like to see more physiological relevance. Reviewer 4 has only textual suggestions. While proving physiological relevance is not strictly required for a Short Report, we agree with the rest of the reviewer concerns and would require some additional experimental revisions to address them, as we consider that this would strengthen the work.

Given the extent of revision needed, we cannot make a decision about publication until we have seen the revised manuscript and your response to the reviewers' comments. Your revised manuscript is likely to be sent for further evaluation by all or a subset of the reviewers.

**IMPORTANT - SUBMITTING YOUR REVISION**

*Re-submission Checklist*

*Published Peer Review*

*PLOS Data Policy*

*Blot and Gel Data Policy*

Sincerely,

Melissa

Melissa Vazquez Hernandez, Ph.D.

Associate Editor

PLOS Biology

REVIEWERS' COMMENTS:

Reviewer #1: 

The work by Loiseau et.al presented an interesting discovery that the periplasmic chaperone Spy is susceptible to chlorite-induced oxidative damage, and the MsrPQ system is responsible to maintaining Spy's activity by reducing its oxidized methionine residues modified under chlorite stress. Overall, the findings are novel, the experiments were well designed, and high-quantity data were obtained. However, I am not fully convinced with the conclusion that Spy is the key target of chlorite-induced oxidative damage and would suggest modifications to further strengthen the paper. 

Major points:

1. I have no doubt that Spy's activity will be affected to some extent by methionine oxidation, but will question on the significance of this modification. The major conclusion were drawn from Spy activity assay where glutamine substitutions were used to mimic methionine sulfoxide. I have to admit that this is indeed a frequently used strategy, however, glutamine with additional amide group is different in polarity so it is possible that this substitution decreases Spy's activity mainly through perturbing the normal 3-dimensional folding of the protein, especially when these substituted methionine residues were within the hydrophobic patches on Spy's concave surface. In other words, I am afraid that the effect of methionine oxidation in Spy might be exaggerated with glutamine substitutions. Essential point: I would suggest the authors to perform the activity assay with Spy purified from the msrP-deletion strain treated with chlorite, or oxidize Spy in vitro using chlorite, and then assay its activity change.

2. The authors demonstrate that the methionine residues on the disordered N-terminus were less critical to affect activity, while Met69, Met87, and Met108 are critical for Spy's chaperone activity. I wonder whether they investigated other conserved methionine residues also located on the concave surface of Spy, namely Met 76, 116, and 120. I bet substituting these residues also had some effect since they also localized in or near the hydrophobic patches. But even with this information, whether these residues were indeed the modification targets under chlorite stress is still in doubts. Is it possible that the authors provide direct evidence which methionine residue(s) in Spy are indeed oxidized upon chlorite treatment, for example, through mass spectrometry characterization? 

3. The authors showed that MsrP is responsible to reverse the oxidation of Spy in vivo by the motility-shift assay in SDS-page, however, it is still unclear whether MsrP help to recover Spy's chaperone activity. Could the authors demonstrate whether the presence of absence of MsrP affect Spy's in vivo activity under chlorite stress, or with purified proteins, whether MsrP recovery the activity of chlorite-oxidized Spy?

Minor points: 

1. The numbering of the methionine residues was counted from the first residue including the signal sequence, but the Q100L is counted from the mature sequence of Spy. This disagreement should be solved.

2. Molecular weights markers should be included in western blots. 

Reviewer #2 (Jan-Ulrik Dahl): 

The bacterial periplasm is extremely vulnerable to a variety of environmental stressors, including oxidative stress caused by reactive oxygen and chlorine species (RO/CS). Previous work by the Ezraty crew nicely demonstrates that reduction of chlorate into chlorite induces the expression of MsrPQ, a periplasmic methionine sulfoxide reductase system that repairs oxidized methionine residues in periplasmic proteins. In this follow-up study, the authors now provide novel mechanistic details on the redox-regulated activity of the periplasmic chaperone Spy. Using proteomic and transcriptomic approaches, Loiseau et al. show that chlorite stress (induced either by chlorite addition or through chlorate reduction) elicits increased Spy expression, a response likely being regulated by the BaeSR TCS rather than by CpxR. The study further provides convincing evidence that Spy's oxidation, particularly of three methionine residues in its "cradle" region, impairs its chaperone activity and that MsrPQ helps restore Spy's functionality by reducing its oxidized methionine, ensuring periplasmic protein quality control. 

The paper is straightforward and easy to read. All experiments were designed and conducted in a logical way with important controls included. I only have minor comments for the authors to consider:

1) The visualization of the proteomic data in Fig 1A (and the supplementary data set) could be improved to make them more accessible to a broader audience. For instance, all significantly up- and downregulated proteins (i.e. >2-fold change) could be color-coded based on regulons and the protein name provided. Do the differentially expressed proteins fall into specific functional groups of interest, e.g. the BaeSR regulon?

2) Given that Spy was the most upregulated protein in the proteomic study, can the authors comment on the oxidation state of three crucial Spy methionine residues?

3) Are the western blots representatives of several biological replicates and did those resulted in the same outcomes?

4) Lines 160-162: please revise to make clear that this is a triple mutant

5) Fig 3 & 5: The spot-titers represent one biological replicate. How many were performed with similar outcomes? Could these data be quantified to consider variability and potential errors in dilution. See for instance triple mutant + pSpyM69Q,M87Q, where the 10-fold dilution gives a full spot but the 100-fold dilution not a single colony.

Kudos to the authors, I really enjoyed reading this manuscript, which adds important information to our understanding of periplasmic protein quality control. This work emphasizes Spy as a critical target of chlorite-induced oxidative damage and underscores MsrPQ's essential role in maintaining chaperone function, protecting its integrity under RCS stress. Well done!

Reviewer #3: 

This study explores the response of Escherichia coli to chlorite-induced oxidative stress, focusing on Spy and its regulation by methionine sulfoxide reductase (MsrPQ). Proteomic analyses show that Spy is the most overproduced protein during chlorite stress. Spy expression is specifically induced by chlorite via the BaeRS stress-response pathway, distinguishing it from responses to other reactive species. The authors propose that chlorite oxidizes methionine residues in Spy, impairing its molecular chaperone activity, but this oxidation is reversible through MsrPQ. Mutational analyses suggest that while N-terminal methionines are non-essential, cradle methionines (Met69, Met87, Met108) are crucial for Spy's function. Methionine oxidation mimics (Met-to-Gln mutants) result in loss of function, phenocopying Spy-deficient strains. In vivo and in vitro assays also imply that oxidized Spy cannot stabilize substrates or prevent protein aggregation, suggesting the critical role of cradle methionines.

This study identifies a chlorite-specific stress response mechanism in E. coli, where MsrPQ may preserve Spy's function by reversing methionine oxidation. However, several issues must be addressed to substantiate the conclusions. 

Major Issues:

1. The band shift observed in SDS-PAGE suggests Spy oxidation in the absence of MsrPQ, but this is insufficient. Methionine oxidation in Spy should be confirmed using peptide mapping analysis (LC-MS/MS).

2. While mutational analyses suggest cradle methionines are crucial, the study lacks data on the activity of oxidized Spy (e.g., in vitro anti-aggregation assays using H₂O₂-oxidized Spy with substrates like MDH and citrate synthase, avoiding reducing agents). Aggregation of these proteins can be induced through temperature shifts or urea dilution without DTT (e.g., Nat Struct Mol Biol. 2011, (3):262-9; J. Biol. Chem. 2020, 295(42) 14488-14500). Additionally, it is interesting to investigate whether MsrPQ can recover Spy activity in vitro.

3. The physiological significance of Spy in response to reactive chlorine species (RCS) remains unclear despite evidence of Spy induction and its role in antibiotic resistance in a Δskp ΔfkpA background. Further exploration and data of its specific roles under RCS stress is needed.

Specific Comments:

1. Title and Key Claims (Lines 138-139, 207-209): These statements are too definitive given the lack of direct evidence for methionine oxidation in Spy under RCS stress.

2. Lines 156-158: Previous studies (e.g., mBio. 2021, 12(5):e0213021) reported observable phenotypes upon spy deletion. Why were no such phenotypes detected or examined in this study?

3. Lines 158-160: Provide additional context to explain the rationale for using this method.

4. Figures 1B, 1C, and 3: Clarify if SDS-PAGE was performed under reducing conditions. Was the Spy band shift reversed by DTT or purified MsrPQ? Include housekeeping proteins as loading controls (e.g., FtsZ for whole-cell lysates, MBP for periplasmic fractions).

5. Figures 5 and 6: Confirm and report whether wild-type and mutant Spy proteins are expressed at equivalent levels using immunoblotting.

6. Line 32: Replace "reveal" with "suggest."

7. Line 33: "Demonstrate" is too strong; consider revising.

8. Replace "chaperones" with "molecular chaperones".

9. Lines 103, 105, 108: Correct figure references to Fig. 1C.

10. Line 190: Specify "penicillin V."

11. Lines 310, 327: Indicate if reducing agents (DTT, β-mercaptoethanol) were included in buffers.

12. Lines 345-346: Correct to "0.2 M Tris-HCl, 1 M sucrose, 1 mM EDTA."

13. Lines 344-349: Provide details on immunoblotting for the periplasmic fraction.

Reviewer #4: 

This short report contains a straightforward set of simple but carefully done experiments that demonstrate that the periplasmic chaperone Spy is both induced by chlorite stress and is susceptible to methionine oxidation by chlorite. As chlorite damages many proteins leading to their inactivation and aggregation it seems reasonable that a chaperone would be induced by a protein damaging agent and that since the Spy chaperone itself is a protein that it is inactivated by chlorate seems reasonable. These results should spur future research. For instance the observation that Spy is cross linked by chlorate is intriguing as Spy does not contain any cysteine residues and I am unaware of methionine modification in a way that would crosslink proteins. This result should be highlighted more in the paper, do their mass spec experiments reveal any cross linked peptides?

I have a few suggestions for improvements:

The precise basis of the phenotypic screen should be made more clear for the naive reader.

Supplementary table 1 needs a lot of work to make it user friendly. For protein validation they required "At least 2 peptides (best hit) and at least one being specific" what exactly does this mean? 

What are the # peptide, # specific peptide and # total peptide columns exactly, they don't seem to add up.

The color scheme is not at all clear, which proteins are in which color classes? Many of the column headings are not immediately clear. Because of these issues It is difficult to determine what the unlabeled proteins in the upper right portion of the figure 1A are, ie those that are induced strongly in fold change and that have significantly p values. Spy though strongly induced in fold is barely above the threshold p value line. It would be good to label about 10 more proteins in this quadrant and in general make supplementary table 1 much more user friendly. 

An indication of the fold overproduction of Spy protein by the plasmid overproducer should be stated.

In the discussion, I think that postulating the N terminal methionine are oxidant sponges is one possibility but a simpler interpretation is that as non-conserved residues they are simply not important for Spy function.

One source of chlorate and chlorite is anthropogenic but the related hypochlorous acid is an key component of the host immunes system of mammals as neutrophils release high concentrations of hypochlorous acid when microbial invasion occurs, this biological source should be mentioned.

---

## [Decision Letter · Decision Letter 2]

15 Aug 2025

Dear Ben,

Thank you for your patience while we considered your revised manuscript "Maintaining the methionine residues of the chaperone Spy in a reduced state is crucial for periplasmic proteostasis." for consideration as a Short Reports at PLOS Biology. Your revised study has now been evaluated by the PLOS Biology editors, the Academic Editor and the original reviewers. 

As you will see in the reports, all reviewers recognize the good job done addressing their previous concerns, but Reviewers #1 and #3 still would like some aspects to be addressed. Reviewer 1 is happy with the revision but requests that the authors clarify some points in the text. Reviewer 3 still thinks that there remain important issues regarding the lack of evidence of Met oxidation under no stress conditions suggesting a new mutant strain, the possible unequal protein loading in the gels, and says that the authors did not properly address his comments 12 and 13 satisfactorily.

IMPORTANT: we have discussed the points raised by Reviewer 3 with the Academic Editor, one point would constitute CBB staining of the gel to confirm equal protein loading, or showing some similar control. Points 3 and 4 should be simple to address as it involves greater clarity/reporting in the text. However, regarding their point 1, we do not require an additional experiments with another strain. 

While addressing the reviewers' concerns, please also address the following editorial requests:

a) We routinely suggest changes to titles to ensure maximum accessibility for a broad, non-specialist readership, and to ensure they reflect the contents of the paper. In this case, we would suggest a minor edit to the title, as follows. Please ensure you change both the manuscript file and the online submission system, as they need to match for final acceptance:

"Repair of oxidized methionine residues in the chaperone Spy maintains periplasmic homeostasis under chlorite stress in Escherichia coli"

Please supply the numerical values either in the a supplementary file or as a permanent DOI’d deposition for the following figures:

Figure 1AC, 4CD, S1AB

c) Please cite the location of the data clearly in all relevant main and supplementary Figure legends, e.g. “The data underlying this Figure can be found in S1 Data” or “The data underlying this Figure can be found in https://doi.org/10.5281/zenodo.XXXXX”

d) Please provide the raw proteomics data and spectra by uploading them in a repository like Zenodo, or proteomic-specific repositories like PRIDE. You can read about our policies and recommendations for repositories here: https://journals.plos.org/plosbiology/s/recommended-repositories

e) We require the original, uncropped and minimally adjusted images supporting all blot and gel results reported in the Figures 1BDC, 3D, 4B, S2B

We will require these files before a manuscript can be accepted so please prepare and upload them now. Please carefully read our guidelines for how to prepare and upload this data: https://journals.plos.org/plosbiology/s/figures#loc-blot-and-gel-reporting-requirements

f) Per journal policy, if you have generated any custom code during the course of this investigation, please make it available without restrictions. Please ensure that the code is sufficiently well documented and reusable, and that your Data Statement in the Editorial Manager submission system accurately describes where your code can be found. Please note that we cannot accept sole deposition of code in GitHub, as this could be changed after publication. However, you can archive this version of your publicly available GitHub code to Zenodo. Once you do this, it will generate a DOI number, which you will need to provide in the Data Accessibility Statement (you are welcome to also provide the GitHub access information). See the process for doing this here: https://docs.github.com/en/repositories/archiving-a-github-repository/referencing-and-citing-content

**IMPORTANT - SUBMITTING YOUR REVISION**

*Resubmission Checklist*

*Published Peer Review*

*PLOS Data Policy*

*Blot and Gel Data Policy*

Sincerely,

Melissa

Melissa Vazquez Hernandez, Ph.D.

Associate Editor

PLOS Biology

REVIEWERS' COMMENTS:

Reviewer #1: 

The authors have made a commendable effort to address my previous concerns by providing additional experimental evidence. The conclusions are now significantly strengthened. In principle, I believe the manuscript is acceptable for publication. However, a few minor points should still be addressed:

1. In lines 213, 218, 220, and 221, the mutant "M46Q/M68Q/M64Q" appears to be a typographical error. Shouldn't the correct mutant be "M46Q/M64Q/M85Q"? Additionally, in Figure 4B, M85Q is mistakenly labeled as M87Q in the western blot and should be corrected.

2. I appreciate the inclusion of mass spectrometry data on the oxidation status of methionine residues in Spy. However, I am still curious about the nature of the identified modifications, whether they are methionine sulfoxide or methionine sulfone. It would be helpful to clarify the nature of the oxidation products. Currently, "methionine sulfoxide" is only mentioned in the abstract. It would improve clarity to define the specific oxidation type in the results section as well.

3. For Figure 4, please provide a reference for the "complete electron donor system" and briefly explain its components or function, as this term may not be familiar to all readers.

4. Also in Figure 4, H₂O₂ was used to induce methionine oxidation. A brief explanation of why H₂O₂ was chosen instead of chlorite would be helpful for readers. Is it due to greater controllability or a more established methodology? Does it produce a comparable oxidation outcome to chlorite?

Reviewer #2: 

The authors have responded well to all feedback. I recommend acceptance and applaud this beautiful study 

Reviewer #3: 

The authors have addressed most of my initial comments; however, several important issues remain and should be considered for further revision:

1. The authors have added new MS-based data quantifying methionine oxidation at each residue in the wild-type and ΔmsrP strains (Fig. 1E). While these results confirm both site-specific Met oxidation and the antioxidative function of MsrP, the effect of chlorite stress on Met oxidation remains unclear, as no data are presented for the non-stress condition. If detecting Met oxidation in Spy Met residues is technically challenging in the absence of stress, a baeS R416C mutant strain—which constitutively activates the Bae pathway—could be a valuable tool to address this point.

2. In the immunoblots of Spy variants, it is unclear whether monomers or dimers are being shown, as molecular weight markers are not indicated. Additionally, the results in Fig. 4B appear inconsistent with those in Figs. 3D and S2. Fig. 4B also lacks a loading control. The use of non-specific protein bands as loading controls is questionable, as their expression may vary under different culture conditions or genetic backgrounds. For example, in the second lane from the left in Fig. 3D, the intensity of the non-specific band is noticeably reduced compared to the others, suggesting either a biological effect or a loading inconsistency. If antibodies against housekeeping proteins are not available, CBB staining of the SDS-PAGE gel can serve as an alternative method to confirm equal protein loading.

3. The responses to my original comments #12 and #13 are not satisfactory. The authors state that periplasmic fractions were used for the immunoblots (Lines 90 and 602), but this does not fully address the concern raised.

4. Lines 227-229 should be revised for accuracy. Although Spy-ox showed reduced activity, it still exhibited clear suppression of aggregation (Fig. 4C). The current description does not adequately reflect the data.

---

## [Editor Report · Decision Letter 3]

11 Sep 2025

Dear Ben,

Thank you for the submission of your revised Short Reports "Repair of oxidized methionine residues in the chaperone Spy maintains periplasmic proteostasis under chlorite stress in Escherichia coli." for publication in PLOS Biology. On behalf of my colleagues and the Academic Editor, Victor Sourjik, I am pleased to say that we can in principle accept your manuscript for publication, provided you address any remaining formatting and reporting issues. These will be detailed in an email you should receive within 2-3 business days from our colleagues in the journal operations team; no action is required from you until then. Please note that we will not be able to formally accept your manuscript and schedule it for publication until you have completed any requested changes.

PRESS

Sincerely, 

Melissa

Melissa Vazquez Hernandez, Ph.D., Ph.D.

Associate Editor

PLOS Biology
